# CONDIFF: A CHALLENGING DATASET FOR NEURAL SOLVERS OF PARTIAL DIFFERENTIAL EQUATIONS

## ABSTRACT

We present ConDiff, a novel dataset for scientific machine learning. ConDiff focuses on the parametric diffusion equation with space dependent coefficients, a fundamental problem in many applications of partial differential equations (PDEs). The main novelty of the proposed dataset is that we consider discontinuous coefficients with high contrast. These coefficient functions are sampled from a selected set of distributions. This class of problems is not only of great academic interest, but is also the basis for describing various environmental and industrial problems. In this way, ConDiff shortens the gap with real-world problems while remaining fully synthetic and easy to use. ConDiff consists of a diverse set of diffusion equations with coefficients covering a wide range of contrast levels and heterogeneity with a measurable complexity metric for clearer comparison between different coefficient functions. We baseline ConDiff on standard deep learning models in the field of scientific machine learning. By providing a large number of problem instances, each with its own coefficient function and right-hand side, we hope to encourage the development of novel physics-based deep learning approaches, such as neural operators, ultimately driving progress towards more accurate and efficient solutions of complex PDE problems.

## 1    INTRODUCTION

In recent years, machine learning techniques have emerged as a promising approach to solving PDEs, offering a new perspective in scientific computing. Machine learning algorithms, especially those based on neural networks, have demonstrated success in approximating complex functions and physical phenomena. Neural networks can provide more efficient and scalable methods compared to traditional numerical methods, which can be computationally expensive and limited by the dimensionality of the problem to be solved. Approaches using physical losses (Karniadakis et al., 2021), operator learning (Li et al., 2020), symmetries incorporation (Wang et al., 2020), data-driven discretization (Bar-Sinai et al., 2019) lead to more physically meaningful solutions and gave neural networks better recognition than just black-boxes.

Classical methods for solving PDEs have been extensively developed and refined over the years, providing a basis for understanding and analyzing various physical phenomena. These methods involve discretization the PDEs using techniques as the finite difference method (LeVeque, 2007), finite element method (Bathe, 2006), finite volume method (Eymard et al., 2000) or spectral methods (Trefethen, 2000), followed by numerical solution of the resulting algebraic equations. While these methods have been successful in solving a wide range of PDEs, they often face the curse of dimensionality when parametric PDEs need to be solved in connection with optimization, optimal control, parameter identification, uncertainty quantification. The reduction of complexity for such classes of problems can be addressed with surrogate models using machine learning.

The main approaches in scientific machine learning are (i) using governing equations as loss functions with physics-informed neural networks (Karniadakis et al., 2021; Cai et al., 2021; Eivazi et al., 2024; Raissi et al., 2019); (ii) learning mappings between infinite-dimensional function spaces with neural operators (Li et al., 2020; Fanaskov & Oseledets, 2023; Lu et al., 2021a; Li et al., 2024; Tran et al., 2021); (iii) hybrid approaches where machine learning techniques are incorporated into classical simulations (Brunton & Kutz, 2022; Schnell & Thuerey, 2024; Hsieh et al., 2019; Ingraham et al., 2018).

These surrogate models have shown significant potential in solving parametric PDEs, but a critical aspect of their development remains the availability of comprehensive datasets for validation. The accuracy and reliability of these machine learning-based approaches are highly dependent on the quality and diversity of the data used to train and test them. Without such datasets, the performance and generalization ability of these models cannot be adequately assessed, and their applicability to real-world problems may be limited. As new techniques and methods emerge in the future, the need for robust and extensive datasets will only increase. It is therefore essential to develop approaches to the curation of high quality datasets that can support the development and validation of innovative approaches to solving complex problems in different scientific and engineering domains.

Typically, scientific machine learning datasets have a large number of parametric PDEs (Takamoto et al., 2022; Luo et al., 2023; Hao et al., 2023) that have a single example per PDE. With ConDiff (short for Contrast Diffusion) we focused on the idea of providing a large number of different realizations for a single problem - the diffusion equation. Currently, ConDiff consists of a diverse set of diffusion equations with 24 realizations, which can be distinguished by complexity, and results in a total of 28800 samples. We also propose an approach to generating complex coefficients for parametric PDEs that can address real-world problems with a measurable metric of the complexity of the dataset.

The ConDiff dataset is available on the Hugging Face Hub: `https://huggingface.co/datasets/condiff/ConDiff`. The code with ConDiff generation, usage, validation and requirements is available at: `https://github.com/condiff-dataset/ConDiff`.

## 2 CONDIFF

**Motivation** Creating a comprehensive benchmark for classes of parametric PDEs is a particular challenge for the scientific machine learning community. The main challenges in creating a comprehensive dataset are: (i) computational complexity; (ii) storage complexity for the desired dimensions of the discretized PDE and parameter space; (iii) properties of the coefficients and solution functions; (iv) relation to real-world problems. The first and second reasons illustrate a technical bottleneck in the creation of the dataset and are mostly dependent on the hardware and efficiency of the numerical method used. Properties such as coefficient smoothness, discontinuity, spatial variation of the coefficients, variance of the parametric space significantly affect the complexity of the dataset and should be carefully chosen. The solution to parametric PDEs (i.e. the ground truth for the dataset) depends on a number of numerical aspects such as choice of mesh, discretization, numerical algorithm, boundary and initial conditions. Therefore, it is very important to consider every little detail regarding different numerical schemes, PDEs, boundary and initial conditions.

Existing benchmarks and datasets cover different aspects of scientific machine learning for different classes of PDEs and can be divided into several groups. PDEBench (Takamoto et al., 2022), PIN-Nacle (Hao et al., 2023), CFDBench (Luo et al., 2023) have a large number of PDEs with different boundary and initial conditions and different dimensionality and resolution. The best covered area is weather forecasting: SuperBench (Ren et al., 2023), ClimSim (Yu et al., 2024), DynaBench (Dulny et al., 2023), OceanBench (Johnson et al., 2024), ChaosBench (Nathaniel et al., 2024). There are also domain specific datasets with applications to Lagrangian mechanics LagrangeBench (Toshev et al., 2024) and phase change phenomena BubbleML (Hassan et al., 2023). Recently, the Flow-Bench (Tali et al., 2024) dataset with complex geometries was introduced. Worth noting frameworks for differential simulations and general environments for PDEs in scientific machine learning: PDE Control Gym (Bhan et al., 2024), PDEArena (Gupta & Brandstetter, 2022), DiffTaichi (Hu et al., 2019), DeepXDE (Lu et al., 2021b) and $\Phi_{\text{Flow}}$ (Holl et al., 2020).

While all of these datasets contribute significantly to the community, to the best of the authors' knowledge there is no dataset dedicated to the very important class of academic and real-world problems, the class of parametric PDEs with random coefficients. Typically, when a new model is proposed, authors test it with a set of equations with smooth coefficients (Brandstetter et al., 2022; Nguyen et al., 2023; Ripken et al., 2023; Bryutkin et al., 2024). Such coefficients do not allow important classes of industrial applications to be addressed. In section 3 we show that increasing the heterogeneity and contrast in the coefficient function leads to increasing challenges in building accurate surrogate models.

**Problem definition**   Existing benchmarks (Takamoto et al., 2022; Hao et al., 2023; Luo et al., 2023) cover a set of PDEs, both steady-state and time-dependent, with different resolutions and time lengths. In our work, we approach the problem from the other side tacking a fixed parametric PDE and generating a comprehensive set of random coefficients for it. We consider a 2D steady-state diffusion equation:

$$
\begin{aligned}
-\nabla \cdot \big(k(x)\nabla u(x)\big) &= f(x), \text{ in } \Omega \\
u(x)\Big|_{x \in \partial\Omega} &= 0
\end{aligned}
\qquad .
\tag{1}
$$

Note that the equation (1) models not only diffusion, but also steady-state Darcy flow in porous media, steady-state heat conduction, etc. To address certain real-world problems, we use the Gaussian Random Field (GRF) to generate the field $\phi(x)$ (Figure 1) with the following covariance models as functions of distance $d$:

- Cubic:
$$
\text{Cov}(d) = \begin{cases} \sigma^2\Big(1 - 7\big(\frac{d}{l}\big)^2 + \frac{35}{4}\big(\frac{d}{l}\big)^3 - \frac{7}{2}\big(\frac{d}{l}\big)^5 + \frac{3}{4}\big(\frac{d}{l}\big)^7\Big), & d < l \\ 0, & d \geq l \end{cases} .
\tag{2}
$$

- Exponential:
$$
\text{Cov}(d) = \sigma^2 \exp\Big(-\frac{d}{l}\Big).
\tag{3}
$$

- Gaussian:
$$
\text{Cov}(d) = \sigma^2 \exp\Big(-\frac{d^2}{l^2}\Big).
\tag{4}
$$

The correlation length in each dataset is $l = 0.05$ and the complexity of a resulting dataset is controlled by variance $\sigma^2$. The forcing term $f(x)$ is sampled from the standard normal distribution for each sampled PDE in each dataset. The resulting coefficient $k(x)$ is obtained with:

$$
k(x) = \exp\big(\phi(x)\big).
\tag{5}
$$

We propose to measure the complexity of the generated GRF with the global contrast in the field $\phi(x)$:

$$
\text{contrast} = \exp\Big(\max\big(\phi(x)\big) - \min\big(\phi(x)\big)\Big).
\tag{6}
$$

**Complexity grows with variance**   By increasing the variance $\sigma^2$ one can obtain a higher contrast (6) and thus a higher complexity of the PDE. This is a well-known phenomenon in applied numerical analysis and can be easily observed empirically. We illustrate this behaviour with the condition number $\kappa(A)$ of the matrices $A$ obtained with discretization of the equation (1).

In the Table 1 one can observe that increasing $\sigma^2$ leads to a higher condition number $\kappa(A) = |\lambda_{\max}|/|\lambda_{\min}|$ of the discretized differential operator (Capizzano, 2003). The condition number is closely related to the performance of the numerical methods used to solve PDEs (Benzi et al., 2005; Elman et al., 2014). A high condition number indicates that small changes in the input can lead to large changes in the output, making the problem ill-conditioned. This is particularly important in PDEs, where small perturbations can significantly affect the solution. Also, if iterative methods are used to solve the discretized PDE, a larger condition number means a larger number of iterations for unpreconditioned and most of preconditioned iterative methods (Saad, 2003).

**Connection to real-world problems**   All of the above reasoning is done with regard to the frequent occurrence of such tasks in real world (Hashmi, 2014; Massimo, 2013; Carr & Turner, 2016; Oristaglio & Hohmann, 1984; Muravleva et al., 2021), including composite materials modeling, heat transfer, geophysical problems, fluid flow modeling. In Figure 2 one can see a cross section of the

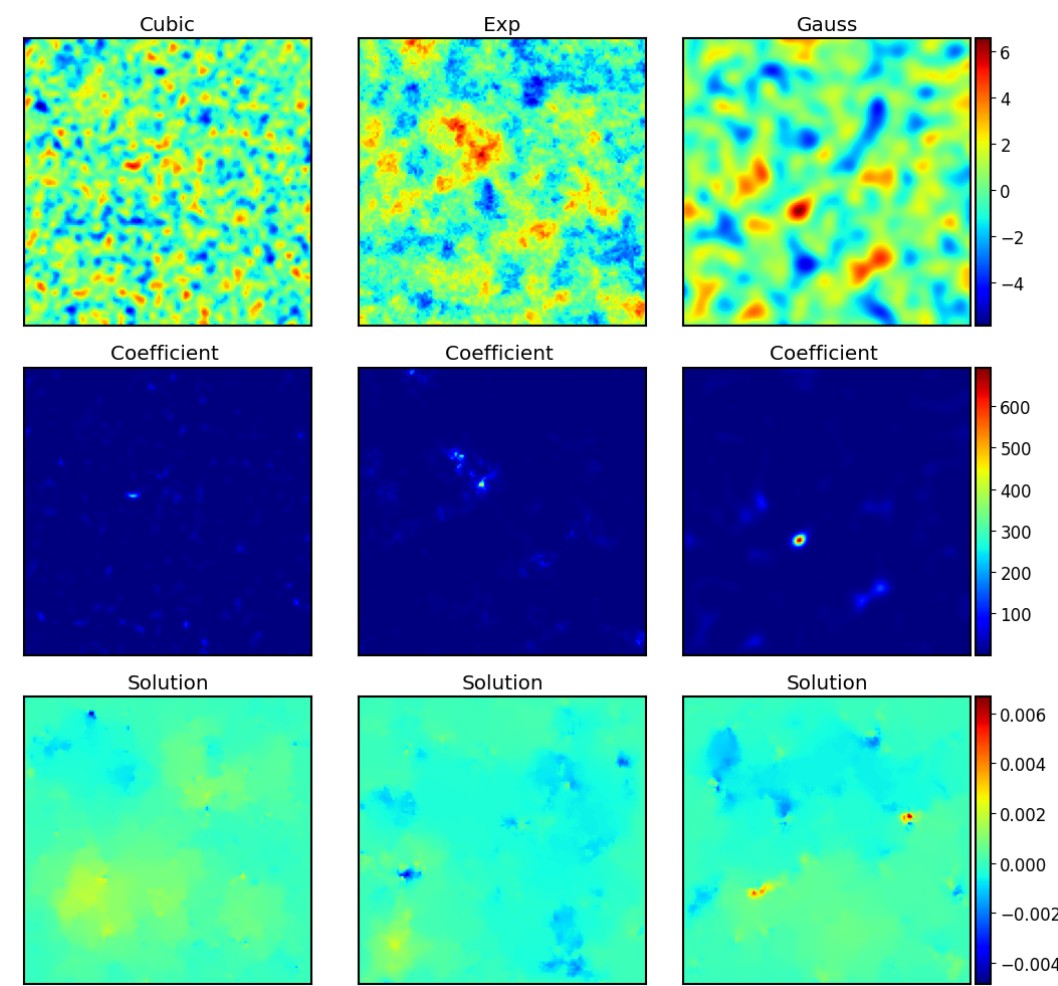

Figure 1: Visualization of the GRF (top row), the coefficient $k(x)$ generated from this GRF (middle row) and the corresponding solution of the equation (1) (bottom row) for a sampled PDEs with grid $128 \times 128$ and $\sigma^2 = 2.0$.

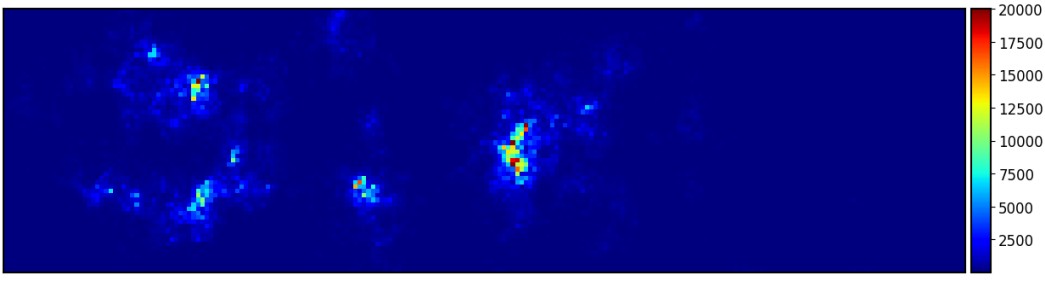

Figure 2: Cross section of the $x-$permeability field along the $z$ axis over the SPE10 model 2 with $z = 4$.

$x-$permeability field along the $z$ axis over the SPE10 model 2 benchmark (Christie & Blunt, 2001). The term permeability is used to denote the coefficients of the above equation when considering flow in porous media. This field is very similar to the ConDiff samples in Figure 1.

This benchmark is well known in the field of reservoir modelling and fluid flow in porous media. SPE10 model 2 poses a significant challenge for the tasks of uncertainty quantification, upscaling and multiphase fluid flow modelling.

Table 1: Summary of the ConDiff with min, mean and max values of the contrast (6). [1]Condition number $\kappa(A)$ is calculated for a single sampled discretized (1).

| Covariance | Variance | Min contrast | Mean contrast | Max contrast | $\kappa^1(A)$ |
|---|---|---|---|---|---|
| | | | Grid $64 \times 64$ | | |
| Cubic | 0.1 | $7.0 \cdot 10^0$ | $1.0 \cdot 10^1$ | $1.5 \cdot 10^1$ | $3.6 \cdot 10^3$ |
| | 0.4 | $5.0 \cdot 10^1$ | $9.6 \cdot 10^1$ | $2.5 \cdot 10^2$ | $7.3 \cdot 10^3$ |
| | 1.0 | $6.0 \cdot 10^2$ | $8.3 \cdot 10^2$ | $1.0 \cdot 10^3$ | $2.0 \cdot 10^4$ |
| | 2.0 | $8.0 \cdot 10^4$ | $8.9 \cdot 10^4$ | $1.0 \cdot 10^5$ | $1.8 \cdot 10^5$ |
| Exp | 0.1 | $6.0 \cdot 10^0$ | $9.0 \cdot 10^0$ | $1.5 \cdot 10^1$ | $4.3 \cdot 10^3$ |
| | 0.4 | $5.0 \cdot 10^1$ | $8.5 \cdot 10^1$ | $2.3 \cdot 10^2$ | $5.2 \cdot 10^3$ |
| | 1.0 | $6.0 \cdot 10^2$ | $7.9 \cdot 10^2$ | $1.0 \cdot 10^3$ | $1.7 \cdot 10^4$ |
| | 2.0 | $8.0 \cdot 10^4$ | $8.9 \cdot 10^4$ | $1.0 \cdot 10^5$ | $1.9 \cdot 10^5$ |
| Gauss | 0.1 | $5.0 \cdot 10^0$ | $8.0 \cdot 10^0$ | $1.4 \cdot 10^1$ | $4.1 \cdot 10^3$ |
| | 0.4 | $5.0 \cdot 10^1$ | $7.5 \cdot 10^1$ | $2.3 \cdot 10^2$ | $8.1 \cdot 10^3$ |
| | 1.0 | $6.0 \cdot 10^2$ | $7.7 \cdot 10^2$ | $1.0 \cdot 10^3$ | $2.4 \cdot 10^4$ |
| | 2.0 | $8.0 \cdot 10^4$ | $8.9 \cdot 10^4$ | $1.0 \cdot 10^5$ | $8.8 \cdot 10^5$ |
| | | | Grid $128 \times 128$ | | |
| Cubic | 0.1 | $8.0 \cdot 10^0$ | $1.1 \cdot 10^1$ | $1.5 \cdot 10^1$ | $1.6 \cdot 10^4$ |
| | 0.4 | $5.5 \cdot 10^1$ | $1.3 \cdot 10^2$ | $2.5 \cdot 10^2$ | $3.8 \cdot 10^4$ |
| | 1.0 | $6.0 \cdot 10^2$ | $8.8 \cdot 10^2$ | $1.0 \cdot 10^3$ | $1.0 \cdot 10^5$ |
| | 2.0 | $8.0 \cdot 10^4$ | $8.9 \cdot 10^4$ | $1.0 \cdot 10^5$ | $1.2 \cdot 10^6$ |
| Exp | 0.1 | $6.0 \cdot 10^0$ | $1.0 \cdot 10^1$ | $1.5 \cdot 10^1$ | $1.7 \cdot 10^4$ |
| | 0.4 | $5.1 \cdot 10^1$ | $1.1 \cdot 10^2$ | $2.5 \cdot 10^2$ | $3.3 \cdot 10^4$ |
| | 1.0 | $6.0 \cdot 10^2$ | $8.3 \cdot 10^2$ | $1.0 \cdot 10^3$ | $9.7 \cdot 10^4$ |
| | 2.0 | $8.0 \cdot 10^4$ | $8.9 \cdot 10^4$ | $1.0 \cdot 10^5$ | $6.3 \cdot 10^5$ |
| Gauss | 0.1 | $5.0 \cdot 10^0$ | $8.0 \cdot 10^0$ | $1.4 \cdot 10^1$ | $1.8 \cdot 10^4$ |
| | 0.4 | $5.0 \cdot 10^1$ | $7.8 \cdot 10^1$ | $2.5 \cdot 10^2$ | $7.2 \cdot 10^4$ |
| | 1.0 | $6.0 \cdot 10^2$ | $7.7 \cdot 10^2$ | $1.0 \cdot 10^3$ | $1.6 \cdot 10^5$ |
| | 2.0 | $8.0 \cdot 10^4$ | $8.9 \cdot 10^4$ | $1.0 \cdot 10^5$ | $1.5 \cdot 10^6$ |

**Dataset description** To generate the fields $\phi(x)$ we use the highly efficient `parafields` library[1] with C++ backend. We use covariance models from {cubic, exponential, Gaussian} with 4 variance values from $\{0.1, 0.4, 1.0, 2.0\}$. We use the forcing term $f(x) \sim \mathcal{N}(0, 1)$. The standard normal force function is chosen to be more complex than a constant forcing term, but not too complex to distract from the complex coefficients, which is the focus of ConDiff. A Dirichlet boundary condition is set for each coefficient realization since boundary conditions do not contribute significantly to the resulting complexity (Capizzano, 2003). The ground truth solution is obtained using cell-centered second-order finite volume method. The coefficients are in the center of cells, the values are in the nodes.

For each parameter set, we generate 1000 training and 200 test realizations of the diffusion equation (1) on $64 \times 64$ and $128 \times 128$ grids. We provide the train-test split in the ConDiff for fair comparison in future research papers. Note that datasets with the same field parameters but different grid sizes are generated independently and do not represent the same field. The fixed geometry of ConDiff allows PDEs with different fields $\phi(x)$ to be compared without fear that different ge-

---

[1]https://github.com/parafields/parafields

ometries will interfere with a fair comparison across different coefficient functions. To control the complexity of the generated PDEs realizations, we set contrast bounds during generation as follows:

- $\sigma^2 = 0.1$, contrast $\in [5, 15]$,

- $\sigma^2 = 0.4$, contrast $\in [50, 250]$,

- $\sigma^2 = 1.0$, contrast $\in [6 \cdot 10^2, 10^3]$,

- $\sigma^2 = 2.0$, contrast $\in [8 \cdot 10^4, 10^5]$.

In total, ConDiff consists of 24 PDEs with different GRFs and grid sizes. Table 1 summarizes the properties of ConDiff. Figure 3 illustrates the contrast distributions. Coming back to the permeability cross section of SPE10 model 2 (Figure 2), it has contrast $= 2.5 \cdot 10^6$ according to (6). We want to emphasize that although the most complex coefficient of ConDiff is smaller by an order of magnitude compared to the cross section of SPE10 model 2, our experiments show that this coefficient is too complex for the chosen models to predict well.

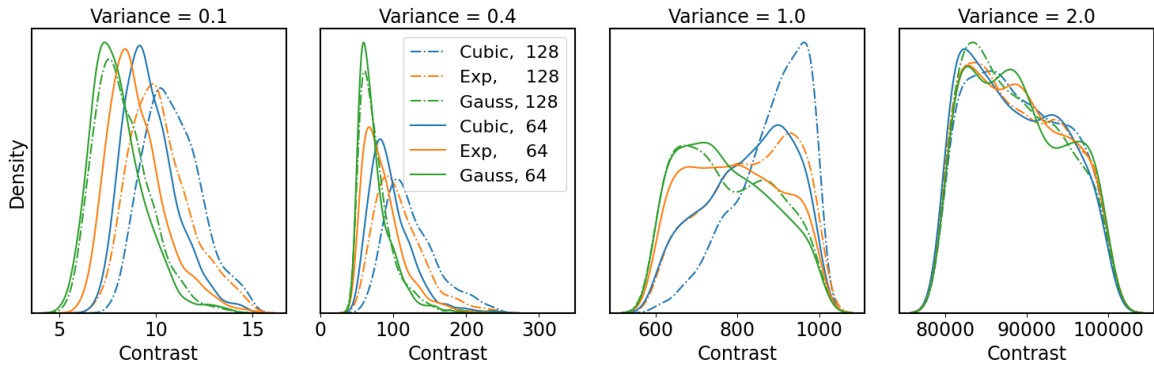

Figure 3: GRF contrast distribution for PDEs from Table 1.

## 3 EXPERIMENTS

**Models** We do not attempt to benchmark every scientific machine learning surrogate model on the ConDiff. Since the ConDiff consists of triplets $\big(k(x), f(x), u(x)\big)$, its primary use is to validate different architectures of neural operators. Therefore, we have selected the following list of models to validate on the ConDiff: Spectral Neural Operator (SNO) (Fanaskov & Oseledets, 2023), Factorized Fourier Neural Operator (F-FNO) (Tran et al., 2021), Dilated ResNet (DilResNet) (Yu et al., 2017) and U-Net (Ronneberger et al., 2015). Neural operators FNO and SNO are both types of neural networks designed to learn mappings between function spaces, in particular to solve PDEs. Neural operators are designed to be universal approximators of continuous operators acting between Banach spaces and to be discretization invariant, meaning that they can handle different discretizations of the underlying function spaces without requiring changes to the model. DilResNet and U-Net are classical neural network models originating from the field of computer vision (CV). Both models have shown their applicability beyond CV and have been used extensively for modeling physical phenomena (Stachenfeld et al., 2021; Ma et al., 2021). More details about the models used can be found in the Appendix A.1.

**Experiment environment** For training neural networks we use frameworks from the JAX (Bradbury et al., 2018) ecosystem: Equinox (Kidger & Garcia, 2021) and Optax (DeepMind et al., 2020). The loss function used is the relative $L_2$ loss:

$$L_2 = \frac{1}{N} \sum_{i=1}^{N} \frac{\|\hat{y}_i - y_i\|_2}{\|y_i\|_2}. \tag{7}$$

Training samples for the models are the values of the coefficient function $k(x)$ and the forcing term $f(x)$ in the grid cells. Targets are the values of the solution function $u(x)$ in the grid cells. We also use (7) as a primary performance metric, assessing the quality of the models' predictions, and report averaged values over the test set with standard deviation.

For all the problems we train for 400 epochs for grid $= 64$ and for 500 epochs for grid $= 128$. We use the AdamW optimizer with an initial learning rate equals to $10^{-3}$ and a weight decay equals to $10^{-2}$. We use a learning rate schedule that halves the learning rate every 50 epochs. Each PDE realization has a dataset size of 1000 training samples and 200 test samples. We use a single GPU Nvidia Tesla V100 16Gb for training on grid $= 64$ and a single GPU Nvidia A40 48Gb for training on grid $= 128$.

Table 2: Results for Poisson equation.

| Grid | SNO | F-FNO | DilResNet | U-Net |
|------|-----|-------|-----------|-------|
| 64 | $0.056 \pm 0.018$ | $0.027 \pm 0.008$ | $0.018 \pm 0.005$ | $0.020 \pm 0.007$ |
| 128 | $0.073 \pm 0.021$ | $0.047 \pm 0.013$ | $0.063 \pm 0.016$ | $0.267 \pm 0.049$ |

Table 3: Performance comparison of the models on the PDEs with the $64 \times 64$ grid from ConDiff.

| Covariance | Variance | SNO | F-FNO | DilResNet | U-Net |
|------------|----------|-----|-------|-----------|-------|
| Cubic | 0.1 | $0.09 \pm 0.02$ | $0.07 \pm 0.02$ | $0.07 \pm 0.02$ | $0.08 \pm 0.02$ |
|  | 0.4 | $0.15 \pm 0.04$ | $0.14 \pm 0.03$ | $0.14 \pm 0.03$ | $0.17 \pm 0.04$ |
|  | 1.0 | $0.23 \pm 0.06$ | $0.22 \pm 0.06$ | $0.22 \pm 0.06$ | $0.24 \pm 0.06$ |
|  | 2.0 | $0.35 \pm 0.10$ | $0.34 \pm 0.09$ | $0.35 \pm 0.10$ | $0.42 \pm 0.10$ |
| Exp | 0.1 | $0.12 \pm 0.03$ | $0.11 \pm 0.03$ | $0.11 \pm 0.03$ | $0.12 \pm 0.04$ |
|  | 0.4 | $0.21 \pm 0.06$ | $0.21 \pm 0.06$ | $0.20 \pm 0.06$ | $0.26 \pm 0.07$ |
|  | 1.0 | $0.33 \pm 0.09$ | $0.34 \pm 0.09$ | $0.36 \pm 0.09$ | $0.35 \pm 0.09$ |
|  | 2.0 | $0.59 \pm 0.14$ | $0.58 \pm 0.14$ | $0.60 \pm 0.13$ | $0.64 \pm 0.13$ |
| Gauss | 0.1 | $0.12 \pm 0.04$ | $0.11 \pm 0.04$ | $0.11 \pm 0.03$ | $0.12 \pm 0.03$ |
|  | 0.4 | $0.23 \pm 0.06$ | $0.22 \pm 0.06$ | $0.21 \pm 0.06$ | $0.25 \pm 0.06$ |
|  | 1.0 | $0.38 \pm 0.08$ | $0.37 \pm 0.09$ | $0.38 \pm 0.09$ | $0.39 \pm 0.09$ |
|  | 2.0 | $0.66 \pm 0.14$ | $0.65 \pm 0.14$ | $0.66 \pm 0.13$ | $0.72 \pm 0.24$ |

Table 4: Performance comparison of SNO and F-FNO on the PDEs with the $128 \times 128$ grid from ConDiff.

| Covariance | Variance | SNO | F-FNO |
|------------|----------|-----|-------|
| Cubic | 0.1 | $0.09 \pm 0.03$ | $0.08 \pm 0.02$ |
|  | 0.4 | $0.15 \pm 0.04$ | $0.14 \pm 0.04$ |
|  | 1.0 | $0.23 \pm 0.06$ | $0.22 \pm 0.06$ |
|  | 2.0 | $0.36 \pm 0.11$ | $0.36 \pm 0.10$ |
| Exp | 0.1 | $0.13 \pm 0.03$ | $0.12 \pm 0.03$ |
|  | 0.4 | $0.21 \pm 0.07$ | $0.21 \pm 0.06$ |
|  | 1.0 | $0.33 \pm 0.09$ | $0.33 \pm 0.08$ |
|  | 2.0 | $0.58 \pm 0.15$ | $0.57 \pm 0.13$ |
| Gauss | 0.1 | $0.13 \pm 0.04$ | $0.12 \pm 0.03$ |
|  | 0.4 | $0.23 \pm 0.06$ | $0.23 \pm 0.06$ |
|  | 1.0 | $0.37 \pm 0.10$ | $0.37 \pm 0.10$ |
|  | 2.0 | $0.68 \pm 0.13$ | $0.66 \pm 0.13$ |

**Validation on ConDiff**   We start the experiments with the Poisson equation and consider it as a special case of (1) with $k(x) = 1$ and contrast $= 1$. All models achieve an accuracy of the order of $10^{-2}$ (Table 2). Increasing the grid size leads to moderate increases in error, except for the U-Net for which the error increases by an order of magnitude.

The diffusion equation for grid $64$ (Table 3) with covariances (2), (3) and (4) are more challenging for the models. While the performance on the diffusion equation with cubic covariance with $\sigma^2 = 0.1$ is comparable to the performance on the Poisson equation, the error on the diffusion equation with exponential and Gaussian covariances is already an order of magnitude higher. Increasing $\sigma^2$ leads to worse performance of each model on each PDE. The most complex PDE is the one generated with the Gaussian covariance model in GRF, which is also consistent with the condition number estimation in Table 1. Interestingly, the performance of FNO and SNO models on PDEs with grid $128$ is not much different from PDEs on grid $64$ (Table 4).

Table 5: Generalization of the models to unseen PDEs with different GRF covariance model with $64 \times 64$ grid and $\sigma^2 = 0.1$.

| | SNO | | | F-FNO | | |
|---|---|---|---|---|---|---|
| Train\Test | Cubic | Exp | Gauss | Cubic | Exp | Gauss |
| Cubic | $0.09 \pm 0.02$ | $0.12 \pm 0.04$ | $0.12 \pm 0.03$ | $0.07 \pm 0.02$ | $0.11 \pm 0.03$ | $0.11 \pm 0.03$ |
| Exp | $0.09 \pm 0.03$ | $0.12 \pm 0.03$ | $0.12 \pm 0.04$ | $0.08 \pm 0.03$ | $0.11 \pm 0.03$ | $0.11 \pm 0.04$ |
| Gauss | $0.09 \pm 0.03$ | $0.12 \pm 0.03$ | $0.12 \pm 0.03$ | $0.08 \pm 0.02$ | $0.11 \pm 0.03$ | $0.11 \pm 0.04$ |
| | DilResNet | | | U-Net | | |
| | Cubic | Exp | Gauss | Cubic | Exp | Gauss |
| Cubic | $0.07 \pm 0.02$ | $0.11 \pm 0.04$ | $0.11 \pm 0.03$ | $0.08 \pm 0.02$ | $0.12 \pm 0.03$ | $0.12 \pm 0.03$ |
| Exp | $0.07 \pm 0.02$ | $0.11 \pm 0.03$ | $0.11 \pm 0.03$ | $0.08 \pm 0.03$ | $0.11 \pm 0.03$ | $0.11 \pm 0.04$ |
| Gauss | $0.17 \pm 0.06$ | $0.25 \pm 0.09$ | $0.11 \pm 0.04$ | $0.08 \pm 0.02$ | $0.12 \pm 0.04$ | $0.12 \pm 0.03$ |

Table 6: Generalization of the models to unseen PDEs with different GRF covariance model with $64 \times 64$ grid and $\sigma^2 = 0.4$.

| | SNO | | | F-FNO | | |
|---|---|---|---|---|---|---|
| Train\Test | Cubic | Exp | Gauss | Cubic | Exp | Gauss |
| Cubic | $0.15 \pm 0.04$ | $0.22 \pm 0.06$ | $0.22 \pm 0.07$ | $0.14 \pm 0.03$ | $0.21 \pm 0.06$ | $0.21 \pm 0.07$ |
| Exp | $0.18 \pm 0.05$ | $0.21 \pm 0.06$ | $0.22 \pm 0.06$ | $0.15 \pm 0.04$ | $0.21 \pm 0.06$ | $0.22 \pm 0.07$ |
| Gauss | $0.17 \pm 0.05$ | $0.22 \pm 0.06$ | $0.23 \pm 0.07$ | $0.15 \pm 0.04$ | $0.21 \pm 0.07$ | $0.22 \pm 0.06$ |
| | DilResNet | | | U-Net | | |
| | Cubic | Exp | Gauss | Cubic | Exp | Gauss |
| Cubic | $0.14 \pm 0.04$ | $0.23 \pm 0.07$ | $0.23 \pm 0.07$ | $0.17 \pm 0.06$ | $0.24 \pm 0.07$ | $0.24 \pm 0.07$ |
| Exp | $0.14 \pm 0.04$ | $0.20 \pm 0.06$ | $0.22 \pm 0.06$ | $0.23 \pm 0.08$ | $0.26 \pm 0.07$ | $0.27 \pm 0.08$ |
| Gauss | $0.30 \pm 0.10$ | $0.24 \pm 0.07$ | $0.21 \pm 0.06$ | $0.21 \pm 0.06$ | $0.27 \pm 0.08$ | $0.26 \pm 0.07$ |

**Transfer between parametric spaces**   Ideally, the surrogate model should handle transfers between different underlying parametric spaces of PDEs without loss of quality. In Tables 5, 6, 7, 8 show that in most experiments the error increases when training on cubic GRF and inferencing on exponential and Gaussian GRF. Conversely, the error decreases when training on Gaussian GRF and inferencing on cubic GRF.

Table 7: Generalization of the models to unseen PDEs with different GRF covariance model with $64 \times 64$ grid and $\sigma^2 = 1.0$.

| | SNO | | | F-FNO | | |
|---|---|---|---|---|---|---|
| Train\Test | Cubic | Exp | Gauss | Cubic | Exp | Gauss |
| Cubic | $0.23 \pm 0.06$ | $0.35 \pm 0.09$ | $0.39 \pm 0.09$ | $0.22 \pm 0.06$ | $0.34 \pm 0.09$ | $0.37 \pm 0.09$ |
| Exp | $0.25 \pm 0.06$ | $0.33 \pm 0.09$ | $0.38 \pm 0.09$ | $0.24 \pm 0.06$ | $0.34 \pm 0.09$ | $0.38 \pm 0.09$ |
| Gauss | $0.24 \pm 0.07$ | $0.35 \pm 0.09$ | $0.38 \pm 0.08$ | $0.24 \pm 0.06$ | $0.35 \pm 0.09$ | $0.37 \pm 0.09$ |
| | DilResNet | | | U-Net | | |
| | Cubic | Exp | Gauss | Cubic | Exp | Gauss |
| Cubic | $0.22 \pm 0.06$ | $0.35 \pm 0.09$ | $0.38 \pm 0.09$ | $0.24 \pm 0.06$ | $0.36 \pm 0.09$ | $0.38 \pm 0.08$ |
| Exp | $0.25 \pm 0.07$ | $0.36 \pm 0.09$ | $0.38 \pm 0.10$ | $0.25 \pm 0.07$ | $0.35 \pm 0.09$ | $0.38 \pm 0.10$ |
| Gauss | $0.57 \pm 0.22$ | $0.59 \pm 0.22$ | $0.38 \pm 0.09$ | $0.27 \pm 0.07$ | $0.36 \pm 0.11$ | $0.39 \pm 0.09$ |

Table 8: Generalization of the models to unseen PDEs with different GRF covariance model with $64 \times 64$ grid and $\sigma^2 = 2.0$.

| | SNO | | | F-FNO | | |
|---|---|---|---|---|---|---|
| Train\Test | Cubic | Exp | Gauss | Cubic | Exp | Gauss |
| Cubic | $0.35 \pm 0.10$ | $0.60 \pm 0.14$ | $0.70 \pm 0.26$ | $0.34 \pm 0.09$ | $0.61 \pm 0.14$ | $0.67 \pm 0.19$ |
| Exp | $0.39 \pm 0.11$ | $0.59 \pm 0.14$ | $0.69 \pm 0.24$ | $0.39 \pm 0.11$ | $0.58 \pm 0.14$ | $0.66 \pm 0.15$ |
| Gauss | $0.40 \pm 0.11$ | $0.60 \pm 0.13$ | $0.66 \pm 0.14$ | $0.37 \pm 0.11$ | $0.60 \pm 0.13$ | $0.65 \pm 0.14$ |
| | DilResNet | | | U-Net | | |
| | Cubic | Exp | Gauss | Cubic | Exp | Gauss |
| Cubic | $0.35 \pm 0.10$ | $0.61 \pm 0.14$ | $0.66 \pm 0.15$ | $0.42 \pm 0.10$ | $0.65 \pm 0.14$ | $0.68 \pm 0.14$ |
| Exp | $0.41 \pm 0.10$ | $0.60 \pm 0.13$ | $0.66 \pm 0.17$ | $0.53 \pm 0.18$ | $0.64 \pm 0.13$ | $0.72 \pm 0.16$ |
| Gauss | $0.72 \pm 0.50$ | $0.68 \pm 0.20$ | $0.66 \pm 0.13$ | $0.66 \pm 0.40$ | $0.69 \pm 0.16$ | $0.72 \pm 0.24$ |

## 4 DISCUSSION

We propose a novel dataset for the field of neural solving of parametric PDEs. The unique feature of the dataset is discontinuous coefficients with high contrast for parametric PDEs from different distributions. By designing the coefficients in this way, we achieve a high complexity of the generated PDEs, which also illustrates real-world problems. The proposed complexity function allows to distinguish between the generated PDEs. We also provide code to generate new data based on the approach used in this paper. Furthermore, we validate a number of surrogate models on the ConDiff to illustrate its usefulness in the field of scientific machine learning.

The practical use of ConDiff is straightforward: it should be used for novel deep learning models and approaches for modeling solution of parametric PDEs from their coefficients. Ultimately, novel deep learning models should exhibit machine-precision prediction quality and not degrade with increasing contrast.

It should be noted that the problems considered in this paper belong to the class of stochastic PDEs. The equation (1) has to be solved for a very large number of sampled coefficients when Monte Carlo or other methods are used to solve the stochastic PDEs. The surrogate models can help to significantly reduce the computational burden, so embedding the surrogate models tested on ConDiff into a Monte Carlo or similar stochastic PDEs solver is a reasonable next step.

## 5 LIMITATIONS

Limitations of the proposed dataset are:

1. For practical numerical analysis, ConDiff is generated with small and moderate variances. The case of large variances has to be studied separately.

2. A linear elliptic parametric PDE is the basis of ConDiff, so other high contrast datasets are needed to test surrogate models for hyperbolic PDEs, nonlinear problems, etc.

3. ConDiff is generated on a regular rectangular grid. Other meshes and geometries may be required as an evolution of ConDiff. This may require more complex computational methods to obtain the ground truth solution.

44. The forcing term $f(x)$ is sampled from the standard normal distributions. While in this paper we focus on the complexity arising from discontinuous coefficients with high contrast, the right-hand side of a PDE can also significantly affect the complexity of the solving PDE. The case of complex forcing terms has to be studied separately.

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

# A APPENDIX

## A.1 ARCHITECTURES

In this section, we discuss the architectures used in more detail and provide information on the training procedures and hyperparameters used. The list of used models is:

1. F-FNO – Factorized Fourier Neural Operator (F-FNO) from (Tran et al., 2021).
2. fSNO – Spectral Neural Operator (SNO). The construction mirrors FNO, but instead of FFT, a transformation based on Gauss quadratures is used (Fanaskov & Oseledets, 2023).
3. DilResNet – Dilated Residual Network from (Yu et al., 2017), (Stachenfeld et al., 2021).
4. U-Net – classical computer vision architecture introduced in (Ronneberger et al., 2015).

**F-FNO** Unlike the original (Li et al., 2020), the authors of (Tran et al., 2021) proposed to changing the operator layer to:

$$z^{\ell+1} = z^\ell + \sigma\left[W_2^{(\ell)}\sigma\left(W_1^{(\ell)}\mathcal{K}^{(\ell)}(z^{(\ell)}) + b_1^{(\ell)}\right) + b_2^{(\ell)}\right],$$

where $\sigma$ is an activation function, $W_1$ and $W_2$ are weight matrices in the physical space, $b_1$ and $b_2$ are bias vectors and

$$\mathcal{K}^{(\ell)}(z^{(\ell)}) = \sum_{d\in D}\left[\text{IFFT}(R_d^{(\ell)} \cdot \text{FFT}_d(z^\ell))\right],$$

where $R_d$ is a Fourier domain weight matrix, FFT and IFFT are Fast Fourier and inverse Fast Fourier transforms.

F-FNO has an encoder-processor-decoder architecture. We used the following parameters: 4 Fourier layers in the processor, 12 modes and GeLU as the activation function. We used 48 features in the processor.

**SNO** We utilized spectral neural operators (SNO) (Fanaskov & Oseledets, 2023) with linear integral kernels:

$$u \leftarrow \int dx A_{ij}p_j(x)(p_i, u)\ ,$$

where $p_j(x)$ are orthogonal or trigonometric polynomials.

These linear integral kernels are an extension of the integral kernels used in the FNO (Li et al., 2020). More specifically, starting from the input function $u^n$, we produce the output function $u^{n+1}$, which is later transformed by nonlinear activation. The transformation depends on the set of polynomials $p_j$ that form a suitable basis for the problem at hand (e.g. trigonometric polynomials, Chebyshev polynomials, etc.). These polynomials are chosen beforehand and do not change during training. The transformation is naturally divided into three parts: analysis, processing, synthesis.

At the analysis stage, we find a discrete representation of the input function by projecting it onto a set of polynomials. To do this, we compute scalar products:

$$\alpha_j = (p_j, u^n) = \int dx p_j(x)u^n(x)w(x)\ ,$$

where $w(x)$ is a non-negative weight function given by the polynomial used.

At the processing stage, we process the obtained coefficients with a linear layer:

$$\alpha_i^{'} = \sum_j A_{ij}\alpha_j\ .$$

Finally, at the synthesis stage, we recover the continuous function as the sum of the processed coefficients:

$$u^{n+1} = \sum_j p_j \alpha_j^{\cdot}.$$

We use SNO in Fourier basis (see (Fanaskov & Oseledets, 2023)) with encoder-processor-decoder architecture. The number of SNO layers is $4$ and the number of $p_j(x)$ is 20. We use GeLU as activation function.

**DilResNet** The conventional dilated residual network was first proposed in (Stachenfeld et al., 2021). In this study, the DilResNet architecture is configured with four blocks, each consisting of a sequence of convolutions with steps of $[1, 2, 4, 8, 4, 2, 1]$ and a kernel size of $3$. Skip connections are also applied after each block and the GeLU activation function is used.

**U-Net** We adopt the traditional U-Net architecture proposed in (Ronneberger et al., 2015). This U-Net configuration is characterised by a series of levels, where each level has approximately half the resolution of the previous one, and the number of features is doubled. At each level, we apply a sequence of three convolutions, followed by max pooling, and then a transposed convolution for upsampling. After upsampling, three more convolutions are applied at each level. The U-Net used in this study consists of four layers and incorporates the GeLU activation function.

