# OpenReview forum: "ConDiff: A Challenging Dataset for Neural Solvers of Partial Differential Equations"
_ICLR.cc/2025/Conference — Submitted to ICLR 2025_

### Official Review · Reviewer_2p4z · 2024-10-27

**Soundness:** 3
**Presentation:** 3
**Contribution:** 1
**Rating:** 3
**Confidence:** 4

**Summary:**

This is a benchmark and dataset paper which presents a new dataset for scientific ML.  ConDiff focuses on the diffusion equation with spatially dependent coefficients, in particular, discontinuous coefficients.   It provides a large number of problem instances, for a single equation, in contrast to other benchmarks which provide fewer instances for many equations.

**Strengths:**

The paper indeed generates benchmark data for the diffusion equation.

**Weaknesses:**

There are already a large number of benchmark dataset for scientific machine learning.
Lines 90-11 mention about 16 of them.  It's not clear how significant another one will be.
This benchmark focuses on just a single equation - but this equation is already in other datasets.  So the novelty is mainly in the choice of parameters of the equation.  This is a rather minor contribution.  The existing  benchmarks aim to be either: very broad, to compare performance of solvers which can handle many different equations, or very applications specific, e.g. specific and realistic physics problems.  The diffusion equation is a foundational equation, but it's is already treated in the benchmarks.  If a user want to dive deeper into this particular equation, it's quite easy to generate solutions using the same methodology as proposed in here: use a numerical solver with a parameterized set of coefficients (e.g. GFF) to generate solutions.

**Questions:**

"For all the problems we train for 400 epochs for grid = 64 and for 500 epochs for grid = 128."
- why not have more grid resolution, e.g. 32, 256, 512?

Limitations:
"The forcing term f(x) is sampled from the standard normal distributions. While in this pa-
per we focus on the complexity arising from discontinuous coefficients with high contrast,
the right-hand side of a PDE can also significantly affect the complexity of the solving
PDE. The case of complex forcing terms has to be studied separately."
- This severely limits the type of solutions, since the forcing term is discontinuous (noise).
Why not also use a GFF to generate smoother forcing terms?  Or use step functions with some geometry, e.g. piecewise rectangles?

---

### Official Review · Reviewer_cFD2 · 2024-11-02

**Soundness:** 2
**Presentation:** 2
**Contribution:** 1
**Rating:** 3
**Confidence:** 4

**Summary:**

This work introduces a dataset called ConDiff for benchmarking operator learning techniques, such as the Fourier Neural Operator (FNO) and DeepONets. The authors consider a two-dimensional diffusion equation of the form $-\nabla\cdot(k(x)\nabla u) = f$ on a square domain with homogeneous Dirichlet boundary conditions and generate different datasets of the form $(k_j,u_j)_{j=1}^N$, where the diffusion coefficients $k_j$ are sampled from Gaussian processes with different covariance kernels. The chosen kernels include cubic, exponential, and Gaussian, with a range of lengthscale parameters.

**Strengths:**

One of the issue in the field of operator learning is the lack of availabiliy of benchmarks, making the comparison between the profusion of neural operator architectures challenging.
This paper aims to address this challenge by introducing a dataset that one can easily use to compare against other techniques.

**Weaknesses:**

There are several weaknesses in this work, summarized in the following points:
1. lack of physical relevance. The authors mention that the class of problems is of "great academic interest" and "shortens the gap with real-world problems". However, they only consider one single simple linear diffusion equation.
2. the authors only consider a simple square domain with rectangular grid. One would typically want to evaluate whether certain architectures can deal with non-periodic boundary condition or non-structured data, which would be closer to real-world applications. This is mentionned in point 3 of the limitations but I do not see why one cannot generate such datasets using a finite element method.
3. In the discussion, the authors state that the "unique feature of the dataset is discontinuous coefficients". However, this has been previously considered in the original FNO paper, and the dataset is publicly available online (containing also more challenging problems), hence the lack of originality of the paper.

**Questions:**

- line 48-62: there seems to be some confusion about the different techniques between PINNs, FNO,... as these have very different purposes (e.g. solving the PDE by minimizing its residual, or learning the solution operator from input-output data pairs).
- line 259: I'm not sure if it's correct to say that the boundary condition does not influence the solution's regularity behavior
- line 262: how do you validate the method, what is the error of the finite volume method? One could use a method of manifactured solution
- line 322: This should be an L^2 loss and include the quadrature weights. In this case it cancels out because of the uniform discretization.
- Tables 2-4: what does that matter?
- line 480: I believe it is neither possible to reach machine-precision accuracy (due to optimization) nor desirable (to avoid overfitting). Additionally, the solution in the benchmark are not resolved to machine precision.

---

### Official Review · Reviewer_twWN · 2024-11-03

**Soundness:** 2
**Presentation:** 2
**Contribution:** 2
**Rating:** 3
**Confidence:** 3

**Summary:**

This paper introduces a benchmark dataset designed to test the capability of machine learning models in solving complex partial differential equation (PDE) systems. The dataset is based on 2D diffusion equation with complex coefficients.

**Strengths:**

The authors propose a novel dataset for neural approaches to solving parametric PDEs, featuring discontinuous coefficients with high contrast across various distributions of parametric PDEs.

**Weaknesses:**

The PDE examined in this paper is relatively straightforward, and existing methods can solve it accurately. Additionally, the models tested on this benchmark are somewhat outdated; for instance, while the authors discuss Physics-Informed Neural Networks (PINNs), they do not include them in the experimental analysis.

The forcing term \(f(x)\) is sampled from a standard normal distribution, keeping the focus on coefficient complexity. However, in many real-world scenarios, the complexity of the forcing term itself significantly affects PDE solutions. Investigating the effects of complex or structured forcing terms in the dataset would make ConDiff more representative of practical challenges and allow for a more comprehensive model evaluation.

The dataset is generated on a regular rectangular grid, which may not fully represent the geometrical complexity found in many real-world applications. Expanding ConDiff to include different grid types or irregular geometries, although computationally demanding, would improve its relevance for practical applications involving complex domains.

**Questions:**

1. Can you test them on recent machine learning models? for example PINN, or Causal PINN[1], and so on.
2. Can you use on more complex PDE system?







[1] Sifan Wang, Shyam Sankaran, and Paris Perdikaris. Respecting causality is all you need for training physicsinformed neural networks. arXiv preprint arXiv:2203.07404, 2022.

**Details Of Ethics Concerns:**

No ethics concerns

---

### Official Review · Reviewer_fiby · 2024-11-04

**Soundness:** 3
**Presentation:** 3
**Contribution:** 1
**Rating:** 3
**Confidence:** 3

**Summary:**

This paper proposes a challenging dataset as a benchmark for models that use neural networks as PDE solvers. The proposed dataset aims to be more representative of real-world data by incorporating discontinuous coefficients with high contrast.

**Strengths:**

Through extensive experiments, the paper demonstrates how the variance changes for each of the four models—SNO, F-FNO, DilResNet, and U-Net—across all train/test scenarios depending on differences in the covariance function.

**Weaknesses:**

First and foremost, I am curious about what differentiates and adds novelty to the dataset proposed in this paper compared to existing datasets created for scientific machine learning (sciML) in previous studies. As cited in the paper, various prior works, such as those by Takamoto et al. (2022), Luo et al. (2023), and Hao et al. (2023), have developed diverse types of datasets as PDE benchmarks. I wonder what major differences exist between this dataset and the ones developed in these studies. The paper mentions that previous datasets lack "a dataset dedicated to the very important class of academic and real-world problems, the class of parametric PDEs with random coefficients." I am interested in understanding more precisely how this differs from previous works. Furthermore, prior datasets include both time-dependent PDEs and steady-state solutions across various resolutions, while this dataset is constructed for a fixed parametric PDE. A more detailed explanation on the novelty of this approach would be helpful.

In relation to the discussion starting on line 159 of page 3, it would be helpful to clarify in more detail how the connection to real-world applications is established. The paper states that Figure 1, showing a dataset created with cubic, exponential, and Gaussian functions using GRF (Gaussian Random Fields), resembles the real-world data shown in Figure 2. However, it would be beneficial to explain more specifically in what ways this dataset offers a unique advantage over other existing datasets. Given the variety of datasets already available, understanding exactly how this proposed dataset provides a novel representation or closer alignment with real-world characteristics would strengthen the paper's argument.

**Questions:**

* In Tables 2–8, while the experimental results are shown for various models across different covariance functions, is there any additional insight that can be provided beyond the raw results? What is the significance of validating models using all three covariance types—cubic, exponential, and Gaussian? Additionally, among the four models tested, can any conclusions be drawn regarding which model performs best?

* Typo in line 498: It seems that "44." is a typo.

* Different Input Scenarios in the Dataset: Were cases considered where both  k(x)  and f(x) are inputs, only k(x) is an input with f(x) fixed, and only f(x) is an input with k(x) fixed? Did the study explore the various cases that could be tested within the proposed dataset?

---

### Comment · Area_Chair_hNMv · 2024-11-25
**Authors' Rebuttal**

Dear Authors,

As the author-reviewer discussion period is approaching its end, I strongly encourage you to read the reviews and engage with the reviewers to ensure the message of your paper has been appropriately conveyed and any outstanding questions have been resolved.

This is a crucial step, as it ensures that both reviewers and authors are on the same page regarding the paper's strengths and areas for improvement.

Thank you again for your submission.

Best regards,

AC

---

### Author Response · Authors · 2024-11-28
**General Rebuttal**

Dear Reviewers,

Thank you for the work you have done in reviewing our paper. We have studied your comments carefully and will be making a major revision of our work.

Regards

---

### Meta-Review · Area_Chair_hNMv · 2024-12-20

**Metareview:**

This paper introduces ConDiff, a new benchmark dataset designed to evaluate machine learning models, particularly neural network-based PDE solvers like FNOs and DeepONets, in solving the 2D diffusion equation with spatially dependent, discontinuous coefficients. It provides a large number of problem instances with varying diffusion coefficients sampled from Gaussian processes with different covariance kernels (cubic, exponential, and Gaussian) and length-scale parameters.

Reviewers questioned the physical relevance of solving such a simple PDE with a trivial square domain, for which traditional methods based on DG and multigrid techniques are already optimal and which ML techniques routinely solve efficiently. Furthermore, it is unclear what added value this dataset offers compared to other benchmarks that also contain 2D diffusion data. Consequently, the contribution of this dataset is minimal; therefore, I recommend rejection.

**Additional Comments On Reviewer Discussion:**

Reviewers mostly agree on the lack of physical relevance, and minimal contribution of the dataset. Authors provided no rebuttal.

---

### Decision · Program_Chairs · 2025-01-22

Reject